# Realization of Phase and Microstructure Control in Fe/Fe_2_SiO_4_-FeAl_2_O_4_ Metal–Ceramic by Alternative Microwave Susceptors

**DOI:** 10.3390/ma15051905

**Published:** 2022-03-04

**Authors:** Chenbo Gao, Pengfei Xu, Fei Ruan, Chenyu Yang

**Affiliations:** School of Materials and Metallurgy, Inner Mongolia University of Science and Technology, Baotou 014000, China; 18168079445@163.com (C.G.); 15201931764@163.com (F.R.); ycy970424@163.com (C.Y.)

**Keywords:** metal–ceramic, microwave sintering, susceptor, phase modulation, nanostructures

## Abstract

This study provides a novel method to prepare metal–ceramic composites from magnetically selected iron ore using microwave heating. By introducing three different microwave susceptors (activated carbon, SiC, and a mixture of activated carbon and SiC) during the microwave process, effective control of the ratio of metallic and ceramic phases was achieved easily. The effects of the three susceptors on the microstructure of the metal–ceramics and the related reaction mechanisms were also investigated in detail. The results show that the metal phase (Fe) and ceramic phase (Fe_2_SiO_4_, FeAl_2_O_4_) can be maintained, but the metal phase to ceramic phase changed significantly. In particular, the microstructures appeared as well-distributed nanosheet structures with diameters of ~400 nm and thicknesses of ~20 nm when SiC was used as the microwave susceptor.

## 1. Introduction

In the traditional industry, the mineral processing process is mostly mining–crushing–beneficiation–smelting–forming, etc., and smelting mainly uses the reduction method, which requires much energy and causes great pollution to the environment. Therefore, it is highly necessary to find a new method to directly prepare mineral powder into composite materials. Considering that metal oxides dominate the minerals, it is expected that these oxides can be reduced using non-polluting means and metal–ceramics can be prepared by adding substances capable of forming ceramic phases.

The preparation of metal–ceramics is mainly based on powder metallurgy [1], chemical deposition [2], and physical deposition [3] and is primarily prepared by adding the metal powder to ceramic raw materials. In addition, the traditional preparation method uses pure materials as the raw material for the preparation of ceramics and uses metals or alloys as the source of the metallic phase. However, ceramics using minerals as the primary raw material are less reported. Since the minerals are mostly metal oxides, which do not directly provide the metals and alloys required to prepare metal–ceramics, they are mainly used for smelting the elements needed. In addition, using minerals to refine metals generates many tailings and environmental pollution problems, so there is an urgent need to find a new process to prepare metal–ceramics directly from minerals to utilize mineral resources and environmental protection comprehensively.

Microwaves are electromagnetic waves with frequencies between 300 MHz and 300 GHz and are initially used in communication [4]. Under microwave irradiation, the electromagnetic field interacts strongly with the wave-absorbing material, causing the directional polarization of the material so that microwaves can be absorbed by the material in a short time to achieve a heating and warming effect. In comparison with conventional heating, microwave heating possesses remarkable features, such as selective heating, internal heating, and non-contact heating [5], which can build a uniform temperature field inside the material and promote the homogenization of the material. At present, microwaves are widely used in the processing or preparing of various materials, such as mineral processing [6], ceramics [7], and metallic materials [8], which have received much attention from scholars at home and abroad. In the process of preparing materials using microwave sintering, with more susceptors to assist in heating [9], commonly used receptors are SiC, activated carbon, graphite, and so on. The dielectric constant of these susceptors usually varies widely with temperature [10,11] and can provide the required heat and also provide some insulation.

However, the use of microwave sintering to convert minerals directly into metal–ceramics is rarely reported, and our team has previously carried out research work on the preparation of new materials using microwave heating of minerals. Based on previous results, this study was conducted to prepare iron-based metal–ceramics by in situ reduction with microwave heating using Baiyun Obo magnetically separated iron ore as the primary raw material. The effect of the regulation of the microwave susceptor on the phase composition, microscopic morphology, and some physical properties of the target products was investigated in detail.

## 2. Experimental Section

### 2.1. Sample Preparation

This experiment used magnetically selected iron ore concentrate from Inner Mongolia as the raw material to prepare ceramic specimens using solid-state synthesis. For the mineral powder, commercial iron ore concentrate powder was used, and its composition was tested by XRF and XRD analysis, as shown in Table A1 and Figure 1. In this study, the mixture used to prepare ceramics contained 85 wt.% iron concentrate, 15 wt.% activated carbon (subsequently referred to as AC), and the kaolinite content was 15 wt.% of the total mass of iron concentrate and AC. Furthermore, the purity of granular AC used in the experiment was >99.90%, the particle size distribution was 0.6–1 mm, and the purity of kaolinite was >99.50%

The final mixture was mixed by ball milling, press molding, and microwave sintering to form ceramic samples with diameters of approximately 25–30 mm and heights of 2–4 mm.

The process used in the experiment mainly consisted of the following. ① Batching. The magnetically separated iron ore concentrate, AC powder, and kaolinite were mixed evenly in proportion, and the mixed raw materials were added to analytically pure alcohol (>99%) as a grinding aid and placed in a planetary ball mill for 24 h at a speed of 480 r/min. ② Drying. We placed the minerals after ball milling into an electric blast box and dried them at 80 °C for 12–24 h until all the grinding aids had evaporated. ③ Sample preparation. After drying, the samples were manually ground in an agate mortar and sieved to achieve a particle size of 74–178 μm. Then, 25-mm-diameter molds were used to compress the samples using a tablet press (pressure of 6 MPa, holding pressure for 1 min). The samples were placed in an isostatic press with a controlled pressure of 220 MPa and held for 90 s. ④ Sintering. The sample after isostatic pressing was heated in a microwave oven, and the specimen was removed after cooling in the oven.

The metal–ceramic samples were sintered by hybrid microwave sintering. A 2.45 GHz microwave heating furnace (DLGR-06S, Zhengzhou DLG Microwave Technology Co., Ltd., Zhengzhou, China) was used for the microwave sintering experiments. A schematic diagram of the hybrid microwave heating system used in this study is given in Figure 2. In this work, the sample was placed in a mullite crucible; three different types of susceptors (AC, SiC, and a mixture of AC and SiC) were selected to assist in heating and to hold the sample; the sample was held for 60 min by microwave irradiation at 2 to 8 kW power input; the local temperature of the sample was measured at 835 °C by a conventional k-type thermocouple, and a schematic diagram of the microwave heating system is shown in Figure 1. Two layers of mullite crucibles were configured around the sample, and the layers were filled with a microwave susceptor to suppress surface heat dissipation and provide an adjustable sintering environment.

### 2.2. Characterization

The samples were analyzed using the PANalytical X ‘Pert powder X-ray diffraction (XRD) spectrometer for physical phase structure analysis (Cu-Kα radiation 40 kV, 30 mA); Raman spectrometry was used to examine the samples, the excitation source was a Nd-YAG laser, excitation wavelength was 532 nm, detection range was 100 cm^−1^–1500 cm^−1^, and data acquisition was performed. The laser intensity was 1–3 mW, the spot size was 2 μm, with a 50× standard objective, and the spectral resolution at room temperature was 0.65 cm^−1^. The sample surface and cross-section were observed by SEM (FE-SEM, Carl Zeiss SUPRA55, Zeiss, Oberkochen, Germany); EDS was used to analyze the sample for elemental analysis.

## 3. Results and Discussion

The raw material used in the sample was magnetically separated iron ore concentrate, the main component of which is Fe_3_O_4_. It has a strong wave absorption capacity and can absorb microwave radiation to achieve self-heating. Magnetite selectively absorbs microwave energy below approximately 650 °C [12].

Figure 3 shows the temperature rise curves collected from experiments without susceptors, reaching a maximum temperature of 545 °C after a heating time of 235 min and failing to reach the reduction temperature of Fe_3_O_4_. The temperature required for the reduction reaction could not be reached because the sample was small and had limited ability to absorb microwave radiation.

The optimum temperature for the thermal decomposition of kaolinite under the action of a microwave field is around 500 °C [13]. Therefore, kaolinite can be partially decomposed in the absence of susceptors. However, Fe_3_O_4_ cannot be reduced due to the low temperature.

A susceptor is a material with high dielectric loss, and its absorbing capacity can change with temperature. When the microwave radiates the susceptor, it can transfer energy to the sample thermally to achieve a heating effect, and once the coupling temperature is reached, the sample can be coupled with the microwave to achieve further heating. Moreover, the susceptor will react with the air inside the furnace to generate a favorable metal oxide reduction atmosphere. Therefore, two types of heating exist throughout the sintering process. 

The main component of the mineral powder used in the experiment was Fe_3_O_4_, one of the simplest ferrites, with excellent dielectric properties. Fe_3_O_4_ is a dielectric-type absorbing material [14,15] as well as a magnetic-loss-absorbing material, and it can absorb microwaves through natural resonance and dielectric loss generated by complex double properties, absorb a large amount of electromagnetic wave energy through eddy current loss, dielectric loss, and ferromagnetic resonance mechanisms, and convert electromagnetic energy into heat to achieve an absorbing effect [16,17], so the frequency dependence is vital; thus, its heat is determined by the loss absorption capacity. The susceptors used in the experiments were AC and SiC, both resistive absorbers. In addition, AC can be subdivided into conductive lossy absorbers, whose absorbing capacity depends mainly on the material’s resistivity.

In the early stage of heating, due to the low temperature in the furnace cavity, the resistance of the susceptor is high at this time, and the mineral powder will be preferred under the action of the microwave; the electromagnetic energy will be converted into heat to achieve the initial heat, which will drive the sample of AC and kaolinite and other components to warm up. With the increase in microwave action time, the susceptor is affected by the heat transfer of the sample; the resistance gradually decreases and begins to self-generate heat.

The sample is heated from the center by microwave radiation, and the external susceptor can heat the surface of the sample, which reduces heat loss and achieves the goal of uniform heating of all parts of the sample. In addition, due to the small size of the sample, the external susceptors can be used as a layer for its insulation. These susceptors reduce the heat loss of the sample and provide the reducing atmosphere and heat source in the closed furnace chamber.

Figure 4 shows the temperature rise curves of different susceptors collected from the experiment; when the microwave susceptor is SiC, the overall average temperature rise rate is 8 °C/min, and it only takes 135 min to reach the target temperature.

When the microwave susceptor is AC, the overall average temperature rise rate is 6 °C/min, it takes 310 min to reach the target temperature, and the temperature rises slowly after reaching 600 °C; when the microwave susceptor is AC/SiC, the overall average rate is between the two, and the temperature rise rate is more moderate. When the microwave susceptor changes to AC/SiC, the overall average rate is between the two, and the warming rate is more sensible. The phenomenon of alternative temperature rise might be ascribed to the dielectric constant difference between AC and SiC, which determined the coupling degree of microwave energy with our samples.

The reaction temperature of the sample can be reached in the presence of the susceptor compared to the temperature rise curve without the addition of the susceptor. The susceptor achieves the insulation of the sample throughout the experiment and enhances the overall warming efficiency. In addition, the dielectric loss of AC and SiC increases with increasing temperature, which improves the overall heating rate [18] and reduces energy consumption.

### 3.1. X-Ray Diffraction Analysis

In order to determine the phase structure of this system, the samples were characterized by XRD after sintering with different microwave susceptors, and the plots are shown in Figure 5. The results showed that the main phase was α-Fe and the ceramic phases were Fe_2_SiO_4_ and minor FeAl_2_O_4_ when the carbon content, kaolinite content, holding time, and sintering temperature were kept constant. The diffraction peaks of the metal and ceramic phases in the samples showed apparent changes with the change in the microwave susceptor. When the microwave susceptor was AC, the sample phase appeared to be mostly a metal phase and a minority ceramic phase; when SiC was used as the microwave susceptor, the sample phase only contained the metal phase α-Fe and a FeAl_2_O_4_ iron–aluminum spinel phase, and the peak intensity of the two diffraction peaks was much weaker than the other two; when the microwave susceptor was AC/SiC, the peak of the ceramic phase became sharp, and the peak intensity of the metallic phase decreased, while their ratio also changed. By comparing different microwave susceptors, it can be concluded that using SiC alone as a microwave susceptor will promote the reduction of α-Fe in the sample but will inhibit the generation of iron olivine phase Fe_2_SiO_4_. In contrast, the homogeneous mixture of AC/SiC can promote the generation of the metal phase and ceramic phase. The ratio of ceramic to metal can be well controlled [19,20].

### 3.2. Raman Spectra

In order to evaluate the structure of the ceramics, Raman analysis was performed on samples prepared with AC as the microwave susceptor. Fe_2_SiO_4_ belongs to an orthorhombic crystal system with a space group of Pbnm, a dense hexagonal stacking of O^2−^, Fe^2+^ occupying octahedral voids, and [SiO_4_] tetrahedra connected through [21,22]. In contrast, FeAl_2_O_4_ belongs to the cubic crystal system with a space group of F_d3m_, with cubic dense stacking of O^2−^, Fe^2+^ occupying tetrahedral voids, and Al^3+^ occupying octahedral voids [23]. Factor group analysis indicates that these two ceramic crystalline phase vibrational modes have significant Raman activity. Based on the above discussion, as shown in Figure 6, the Raman test results reveal consistent Raman characteristic peaks for different microwave susceptor samples: 291 cm^−1^, 406 cm^−1^, 499 cm^−1^, 668 cm^−1^ correspond to 1 Eg and 3 T_2g_ vibrational modes of FeAl_2_O_4_, while 610 cm^−1^ can be attributed to the antisymmetric deformation vibration of O-Si-O in Fe_2_SiO_4_. The low wavenumber 223 cm^−1^ may be related to the bending beat of O-Fe-O, and the high wavenumber 1317 cm^−1^ can be attributed to the second-order Raman peak of the magnetic oxide [24,25]. This corresponds with the XRD results and proves the presence of the ceramic phases Fe_2_SiO_4_ and FeAl_2_O_4_.

### 3.3. Microstructure Analysis

It can be seen from Figure 7 and Figure 8 (MAG:10kX) that the product structure is dense. Under constant holding time, sintering temperature, and kaolinite content, the change of the microwave auxiliary susceptor significantly affects the samples’ grain size and microscopic morphology. When the microwave susceptor is AC, most of the grain sizes are in the range of 2–5 μm, and the EDS results show that the bright white part is the metal Fe matrix, and the dark part is the Fe_2_SiO_4_ iron olivine enhanced phase. The organization of the iron-based phase is distributed in granular form, and there are a small number of microporous defects between the particles, which is related to the complete discharge of CO generated by the carbon reaction in the sample. When the microwave susceptor is SiC, the sample has the most α-Fe content. The iron particles are coarser, up to approximately 5 μm, and the ceramic organization almost wholly disappears due to the role of SiC in the microwave, generating a strongly reducing atmosphere, promoting the transformation of the metal phase, and inhibiting the generation of the ceramic phase. When the microwave susceptor is AC mixed with SiC, the metal phase of the sample and the ceramic phase achieve a synergic effect between the crystals to achieve a close combination of fewer voids and a clear decentralized structure. 

When changing the microwave susceptor to SiC, the morphology and structure of the reaction products changed significantly, as shown in Figure 9. Uniform nanosheets with diameters of ~400 nm and thicknesses of ~20 nm were formed in large quantities in the mineral phase, and their structures were mainly composed of metallic iron. This Fe nanostructure is highly reactive and rapidly transforms into a micrometer columnar crystal structure with Fe_2_O_3_ as the main phase after heat treatment in an electric furnace, with a small amount of ceramic phase precipitated. Notably, when AC/SiC mixtures were chosen as microwave susceptors, the products’ morphology and microstructure were altered. Many nanosheets with a diameter of ~400 nm and thickness of ~20 nm were formed during the microwave process, crystallizing in metallic iron. Fe nanosheets possessed high reaction activity, demonstrated by the fast transformation to a Fe_2_O_3_ micro-columnar structure with a minor ceramic phase after low-temperature treatment in a muffle oven. Therefore, the utilization of alternative susceptors can regulate the metal–ceramic microstructure conveniently.

### 3.4. Heating Rate

To investigate whether the heating rate promoted the formation of such nanostructures, we designed a conventional sintering experiment comparing the microwave heating rates, and the XRD results of the samples are shown in Figure 10.

It can be seen that with the use of conventional sintering, simulating different heating rates of the susceptors, all three different heating rates produced a large amount of Fe_2_O_3_ and only a smaller amount of ceramic phase (Fe_2_SiO_4_). Moreover, the peak intensity of Fe_2_O_3_ and Fe_2_SiO_4_ did not change significantly when comparing different heating rate conditions. 

As can be seen from Figure 11, the structure of the product using conventional sintering is dense. The variation of the heating rate has almost no significant effect on the grain size and microscopic morphology of the samples. None of the experiments simulating the three different temperature rise profiles produced the layered nanosheet structures and nanopillars as in Figure 9a,b. 

### 3.5. Reaction Mechanisms

Under the action of the microwave field, with the occurrence of each reaction, the fluctuation of the oxygen concentration occurred in the closed furnace cavity, which caused the formation of a small amount of FeO in the sample locally, accompanied by the transfer of absorbed energy, which caused a wide separation of the two [26], and the remaining part of FeO was obtained by the reaction of the reducing atmosphere in the furnace cavity with Fe_3_O_4_. As the added kaolinite is an aluminosilicate (Al_2_O_3_-2SiO_2_-2H_2_O), it can be decomposed into Al_2_O_3_, SiO_2_, and water vapor under the action of the microwave, providing a large amount of Al_2_O_3_ and SiO_2_ for the system, which enables the iron oxides to combine with these two preferentially for a related reaction, thus inhibiting the reaction with other impurities in the mineral powder (MnO_2_, etc.) reaction.

The reaction products using AC as a susceptor have been discussed in previous articles [27]. In the process of preparation, FeO is the critical link between metal phase and ceramic phase generation; there are three consumption pathways, and there is competition between the three processes, so the ratio of metal phase to ceramic phase in this system can be adjusted by adding kaolinite and changing the microwave susceptor. The comparison of kaolinite is adjusted by whether the reaction oxides are in complete contact, while the comparison of microwave susceptors is adjusted by the reaction thermodynamics [27].
(1)$$2C+O_{2}=2\mathrm{CO}$$
(2)$$\mathrm{SiC}+2O_{2}={\mathrm{SiO}}_{2}+{\mathrm{CO}}_{2}$$
(3)$${\mathrm{SiO}}_{2}+C=\mathrm{SiO}+\mathrm{CO}$$
(4)$$C+{\mathrm{CO}}_{2}=2\mathrm{CO}$$

When the susceptor is SiC and AC, microwaves generate heat, providing a heat source for the reaction and reducing heat loss. In addition, SiC consumes O_2_ in the furnace, which inhibits the consumption of O_2_ by AC in the sample, hinders Fe’s formation, and promotes FeO’s reaction with free SiO_2_ to form Fe_2_SiO_4_. From the heating curves in Figure 11, it is known that the dielectric coefficients [28] and heating characteristics of the two are different when a hybrid susceptor is used. In addition, it can be found from reactions (1) and (2) that there is competition between AC and SiC for O_2_, which leads to a slow heating rate and a lack of reducing atmosphere, and these factors cause the process of the iron oxide reduction reaction to be slower and instead promote the reaction of generating the ceramic phase.

When the microwave susceptor is SiC, the microwave field can generate the temperature field of the diffusion homogenization system in the furnace cavity so that the water vapor generated by the decomposition of kaolinite races within the pore space, promoting the migration kinetics of the material and providing power for the reduction of Fe, leading to an increase in the metallic phase and a decrease in the ceramic phase. Figure 12 shows the Gibbs free energy versus reaction temperature, from which it can be found that the Gibbs free energy of reaction (2) is negative, except for the positive Gibbs free energy of reactions (3) and (4). These indicate that reactions (3) to (4) are possible in thermodynamic entropy, but higher temperatures are required under conventional conditions, and microwave heating can reduce the reaction temperature, so it is possible to occur under the requirements of this system. When SiC is used as the microwave susceptor, due to the presence of some O_2_ in the furnace, SiC then reacts with it to produce some gaseous SiO_2_ as well as CO_2_, and gaseous SiO_2_ continues to respond with solid C in the sample to make gaseous SiO and CO; the air in the furnace also reacts with solid AC in the sample to produce CO, providing the required reducing atmosphere for the sample [29]. The Gibbs free energy of reaction (2) is lower than the Gibbs free energy of reaction (1), so reaction (2) occurs first, and SiC can react with AC and air to produce CO_2_ first, which can then create a further CO atmosphere for the reduction of iron oxides. Therefore, the initial time for the reduction reaction in the furnace chamber is much earlier when the susceptor is SiC than when the other two susceptors are present. This is why more α-Fe can be produced under this condition.

In addition, the micro-focusing and polarization effects of SiC in the microwave field can also achieve the effect of accelerating the microstructure evolution, leading to more concentrated heat generation, which allows the reaction to proceed rapidly. Under this condition, the iron oxides in the system very easily generate iron–aluminum spinel, which subsequently reacts with free silica to form mullite and, through related reactions, leads to metallic iron. This process is almost consistent with the experimental results of Xiao-bin LI [30], and we did detect the presence of quartz solid solution in XRD, as shown in Figure 11. Since kaolinite is a layered structure, the combination of iron oxides can also generate a similar structure, rapidly reduced under the action of the microwave field and reducing atmosphere; the influencing factors need to be further investigated.

## 4. Conclusions

Fe/Fe_2_SiO_4_-FeAl_2_O_4_ metal–ceramic can be prepared by introducing altered susceptors under microwave irradiation with mineral powder, AC, and kaolinite. This metal–ceramic is rich in Fe_2_SiO_4_, so it can be expected to be used in high-temperature-resistant applications.

(1)The ratios between the metal and ceramic phase and microstructures of the ceramics can be regulated by changing the microwave susceptor. The valuable insight into the interaction of alternative microwave susceptors with samples could provide theoretical and practical guidance for microwave metallurgy in synthesizing advanced materials.(2)Highly active Fe nanosheets can be obtained by modulating microwave susceptors. Inhibiting the oxidation of Fe nanosheets while promoting the precipitation of the ceramic phase is expected to optimize the synthesis of nanocomposite metal–ceramics, which has great potential in the design of structural materials and catalytic materials.(3)In preparing metal nanocomposites, the grain growth morphology can be controlled by suppressing the metal nanosheets’ oxidation and the ceramic phase development.

## Figures and Tables

**Figure 1 materials-15-01905-f001:**
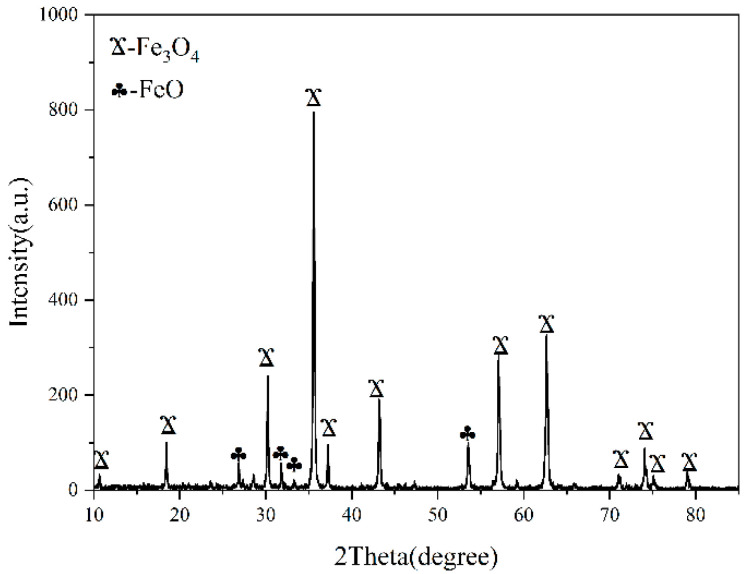
XRD of iron ore concentrate powder.

**Figure 2 materials-15-01905-f002:**
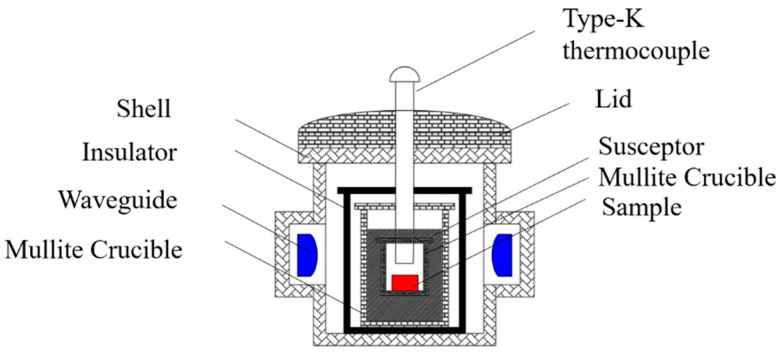
Microwave heating device central part.

**Figure 3 materials-15-01905-f003:**
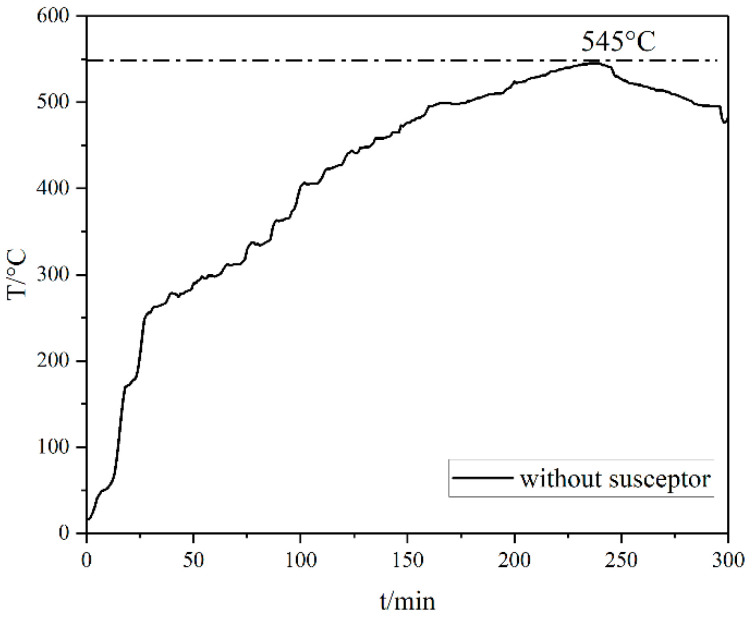
Temperature rise curve without a susceptor.

**Figure 4 materials-15-01905-f004:**
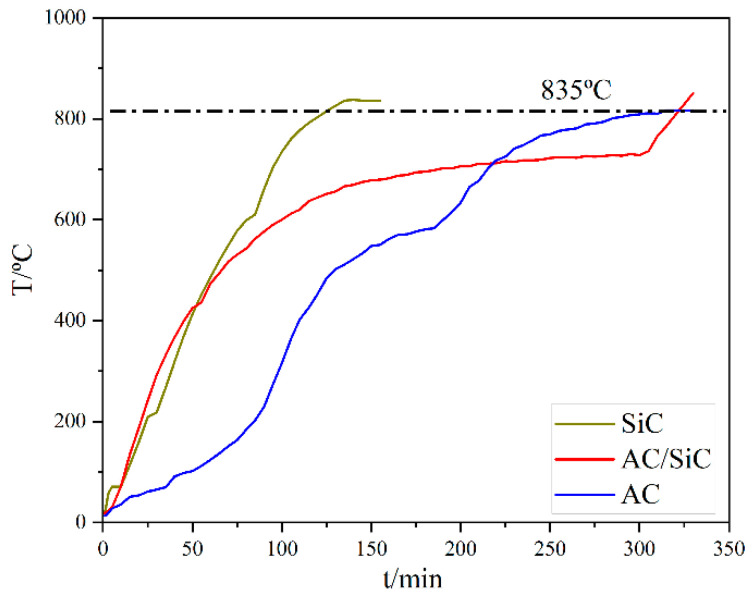
Temperature rise curve of the different microwave susceptors.

**Figure 5 materials-15-01905-f005:**
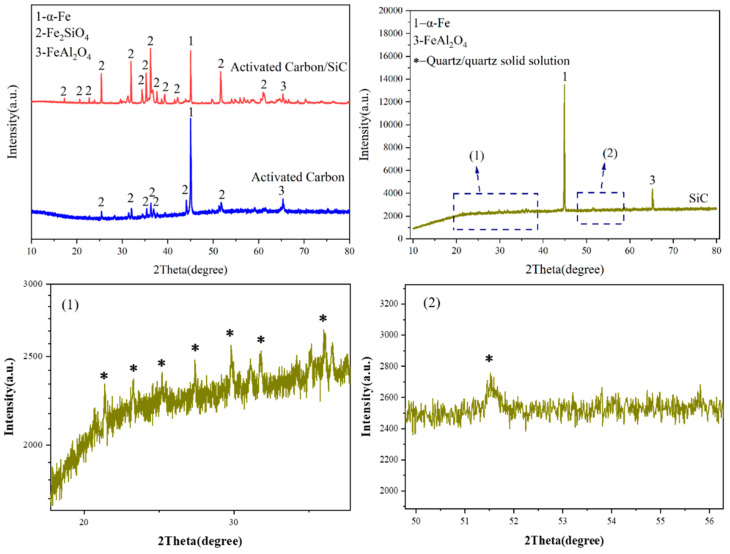
XRD of samples prepared with different microwave susceptors.

**Figure 6 materials-15-01905-f006:**
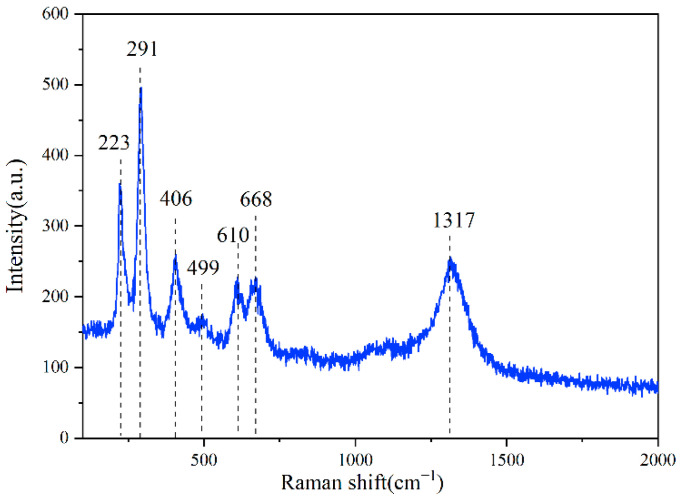
Wave-absorbing susceptor for AC microwave preparation samples.

**Figure 7 materials-15-01905-f007:**
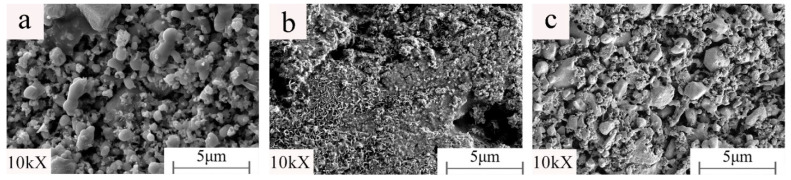
Samples prepared with different microwave susceptors: (**a**) AC; (**b**) SiC; (**c**) AC/SiC = 1.

**Figure 8 materials-15-01905-f008:**
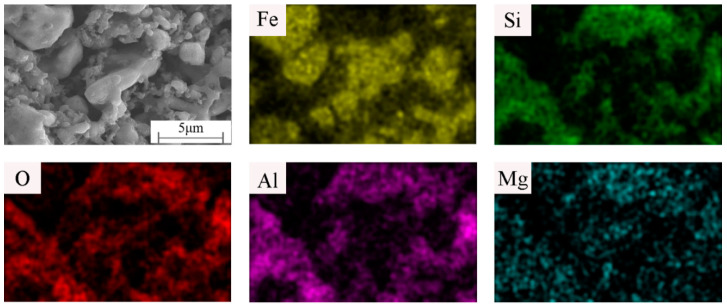
EDS elemental scans of samples prepared with the microwave susceptor as AC/SiC = 1 hybrid.

**Figure 9 materials-15-01905-f009:**
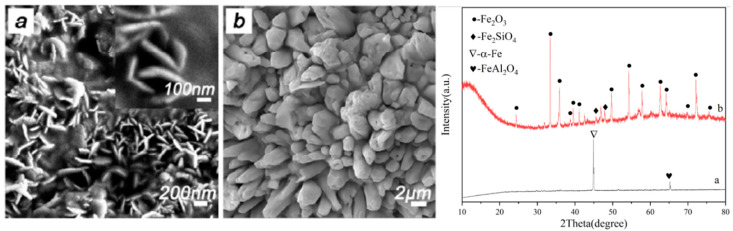
Samples prepared with microwave susceptor for SiC microwave. (**a**) Before heat treatment (**b**) After heat treatment.

**Figure 10 materials-15-01905-f010:**
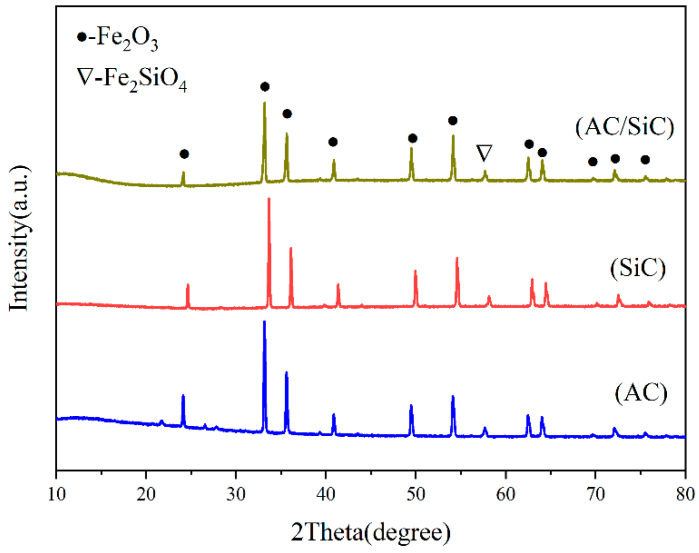
XRD of samples prepared at different heating rates.

**Figure 11 materials-15-01905-f011:**
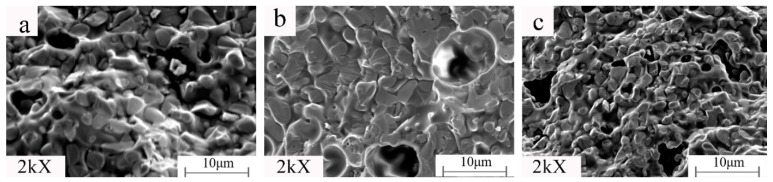
Samples prepared by simulating different susceptors warming rates: (**a**) AC; (**b**) SiC; (**c**) AC/SiC = 1.

**Figure 12 materials-15-01905-f012:**
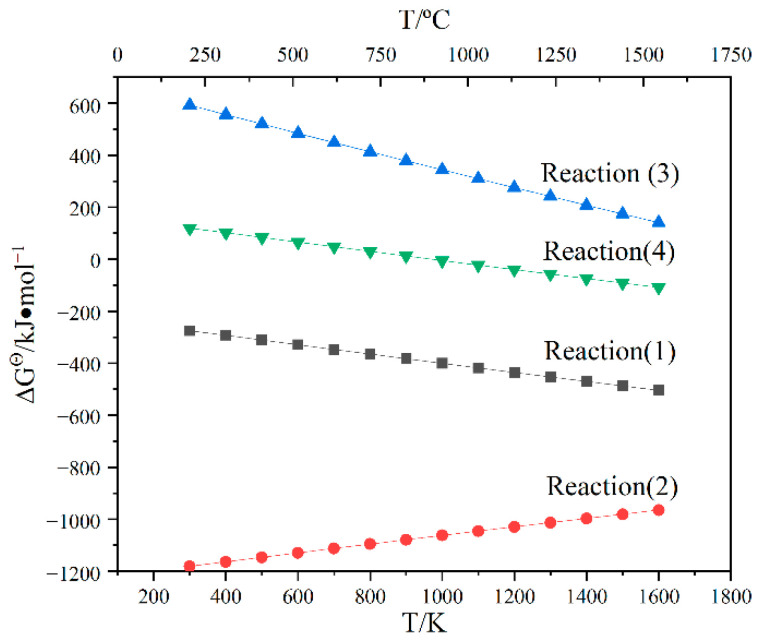
Gibbs free energy versus temperature for chemical reactions.

## Data Availability

Data are contained within the article.

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
