# Peer review of "Realization of Phase and Microstructure Control in Fe/Fe2SiO4-FeAl2O4 Metal–Ceramic by Alternative Microwave Susceptors"

_materials, 2022, doi:10.3390/ma15051905_

Round 1

Reviewer 1 Report

The manuscript "Realization of phase and microstructure control in Fe/Fe2SiO4-FeAl2O4 metal-ceramic by alternative microwave susceptors" was reviewed. The paper is well organized and suggested for publication. Before publication, there some minor comments that need to be addressed:

  1. The title of the manuscript is long. It should be short and informative.
  2. In Fig.1, some of the information are missed. Figure 1 should be replaced by the new one with more details. If possible, provide the image of the used microwave furnace. 

Author Response

Dear Reviewers:
Thank you for your comments concerning our manuscript entitled “Realization of phase and microstructure control in Fe/Fe2SiO4-FeAl2O4 metal-ceramic by alternative microwave susceptors” (ID: materials-1562484). We thank the reviewer for the very helpful comments. We revised the manuscript according to the recommendations. We provide below with a brief “List of changes” in the revised manuscript, followed by our answers to the referee’s comments.

List of changes:

Comment 1: The title of the manuscript is long. It should be short and informative.

Response to Comment 1:

Our main starting point is to prepare waste tailings into ceramic materials by means of microwave treatment. However, the composition of tailings is complicated, and there are few reports on this in our country. So we have first explored the process based on iron ore concentrate and will subsequently extend to the utilization of tailings.

The main purpose of this article was written to represent the research of nanostructures in the course of process investigations, with a focus on changes in the susceptors that caused changes in the microstructure.

As the reviewer said, the current title is too long, although it clearly expresses the central idea of the article. We are exploring the streamlining of the article title, and due to time constraints, the new title will be changed prior to publication.

Comment 2: In Fig.1, some of the information are missed. Figure 1 should be replaced by the new one with more details. If possible, provide the image of the used microwave furnace.

Response to Comment 2:

We have made changes to address the reviewer's comment that less information is expressed in the figures and have adjusted the font.

This is my first time writing an article in English, so I'm sure there are many shortcomings, and I hope the reviewers will criticize and correct me.

We thank the reviewer and remain at your disposal for any further questions.

Yours sincerely,

Chen-bo Gao

Reviewer 2 Report

Authors C.-b. Gao  et all. studied a phase development in the samples when heated an iron ore mixed with activated carbon (AC), SiC and a mixture of both additives.  Since the authors used combining heating regime of microwave heating and conventional heating, these additives had a role of microwave susceptors.

The authors do not explain in the manuscript why they performed the described investigation, because a reader may believe that the iron ore is used as a starting material for manufacture metal Fe. So, what was the purpose of the described study? In the Conclusions section the authors claim that “…ceramics with enhanced properties can be prepared…”. What are these properties? Are there any possible applications for such Fe-ceramic composites?

Pg. 2, line 61: “solid-phase method”. What this does it mean? Is this maybe solid-state synthesis? which is the common method for preparation of ceramics.

Pg. 2, lines 61-63. “The main components of the ore were tested by XRF analysis, and the chemical composition of this ore was obtained, as shown in Table 1,” It is not clear what are the main components of the ore. Are these main components the iron concentrate?

Pg. 2, lines 63-66: “In this study, the mixture used to prepare ceramics contained 85 wt.% iron concentrate, 15 wt.% Activated Carbon (Subsequently all called AC), and the kaolinite content was 15 wt.% of the total mass of iron concentrate and AC.”. This sentence is not clear and it should be improved.

  1. 2, line 66: “particle size is concentrated in 0.6-1 mm”. What does it mean concentrated?

Pg. 2, lines 70-71. Planetary ball mill, presses… are not characterization equipment!

The authors present a composition of iron ore concentrate in Table 1 as a percentage of different oxides. Are there wt%? In the table  there is 65,5% TFe (what is TFe?) and 27.9% FeO. In the pg. 3, line 111 (also pg. 4 line 133), authors claim that the main component of iron ore is Fe3O4.  This is rather confusing. Additionally,  it is strongly suggested, that the authors also present phase composition of the starting iron ore.

Pg. 5, line 190: “weak ceramic phase”. The “weak” is inappropriate term for ceramic phase.

The authors describe microstructure properties of their samples based on the surface of as-prepared samples. Actually, correct analysis of the microstructure can be only done on properly grinded and polished cross-section of the samples. Only such obtained information reflect the bulk properties of the analyzed material.  

Author Response

Dear Reviewers:
Thank you for your comments concerning our manuscript entitled “Realization of phase and microstructure control in Fe/Fe2SiO4-FeAl2O4 metal-ceramic by alternative microwave susceptors” (ID: materials-1562484). We thank the reviewer for the very helpful comments. We revised the manuscript according to the recommendations. We provide below with a brief “List of changes” in the revised manuscript, followed by our answers to the referee’s comments.

List of changes:

Comment 1: What was the purpose of the described study?

Response to Comment 1:

Since the abandoned tailings in our country are not well utilized, they have caused serious pollution to the environment, and the purpose of our research is based on the recycling of resources.

Our main starting point is to prepare waste tailings into ceramic materials by means of microwave treatment. However, the composition of tailings is complicated, and there are few reports on this in our country. So, we have first explored the process on the basis of iron ore concentrate and will subsequently extend to the utilization of tailings.

Comment 2: Pg. 2, line 61: “solid-phase method”. What this does it mean? Is this maybe solid-state synthesis? which is the common method for preparation of ceramics.

Response to Comment 2:

We expected to prepare the iron concentrate directly into ceramics, so we did choose the solid-state synthesis method, and we focused on the process, which is why we used this simpler method.

Since this is my first time writing an article in English, I may have used inappropriate words. After studying, I did use the wrong words, and now I have corrected them. Thank you very much for your criticism and guidance.

Comment 3: Pg. 2, lines 61-63. “The main components of the ore were tested by XRF analysis, and the chemical composition of this ore was obtained, as shown in Table 1,” It is not clear what are the main components of the ore. Are these main components the iron concentrate?

Response to Comment 3:

The composition table of this ore was provided by the manufacturer, and we also performed XRF analysis on it; and the final results presented are the results of our analysis. The identification of pure magnetite is based on the following formula.

Ñ (TFe)/ Ñ (FeO)=2.3

The mineral we used Ñ  (TFe)/ Ñ  (FeO) has a ratio of 2.3, so it is pure magnetite.

Comment 4: Pg. 2, lines 63-66: “In this study, the mixture used to prepare ceramics contained 85 wt.% iron concentrate, 15 wt.% Activated Carbon (Subsequently all called AC), and the kaolinite content was 15 wt.% of the total mass of iron concentrate and AC.”. This sentence is not clear and it should be improved.

Line 66: “particle size is concentrated in 0.6-1 mm”. What does it mean concentrated?

Response to Comment 4:

It is true that this is not clear enough and can be easily misunderstood. We have changed and streamlined it and highlighted it in red in the text.

Comment 5: Pg. 2, lines 70-71. Planetary ball mill, presses… are not characterization equipment!

Response to Comment 5:

I apologize for not being able to proofread the experimental section in detail carefully because I was in a hurry to submit the manuscript, and I did make an oversight in this regard and have corrected it.

Comment 6: The authors present a composition of iron ore concentrate in Table 1 as a percentage of different oxides. Are there wt%?

Response to Comment 6:

This is indeed wt%, due to lack of experience and not having a good grasp of the details, and I have made changes in the text.

Comment 7: In the table there is 65,5% TFe (what is TFe?) and 27.9% FeO. In the pg. 3, line 111 (also pg. 4 line 133), authors claim that the main component of iron ore is Fe3O4.  This is rather confusing. Additionally, it is strongly suggested, that the authors also present phase composition of the starting iron ore.

Response to Comment 7:

TFe is total iron which refers to the total content of elemental iron, and this composition table is generally used in industry to indicate the amount of iron in a mineral. The minerals used in the experiments were obtained after magnetic separation, and the main composition is Fe3O4, which can be seen as FeO·Fe2O3. According to your suggestion, and XRD diagram has been attached to illustrate the main components in the mineral.

Comment 8: Pg. 5, line 190: “weak ceramic phase”. The “weak” is an inappropriate term for the ceramic phase.

Response to Comment 8:

Since the statement in the text is really inappropriate, we have changed it and highlighted it in red.

Comment 9: The authors describe microstructure properties of their samples based on the surface of as-prepared samples. Actually, correct analysis of the microstructure can be only done on properly grinded and polished cross-section of the samples.

Response to Comment 9:

The microstructure pictures in the article are representations of cross-sections, except for EDS, which was examined using the surface of the sample. The lack of clarity caused misunderstanding by the reviewers, which we have corrected in the article.

This is my first time writing an article in English, so I'm sure there are many shortcomings, and I hope the reviewers will criticize and correct me.

We thank the reviewer and remain at your disposal for any further questions.

Yours sincerely,

Chen-bo Gao

Reviewer 3 Report

  1. Summary, strengths, weaknesses, overall contribution

Summary: In the paper the Authors provided a novel method to prepare metal-ceramic composites from magnetically selected iron ore using microwave heating. With the use of XRD and SEM they have investigated the effect of three susceptors on the mictostructure of the composites.

General strengths: The presented method is innovative and may find application in the field of composites production. It is also environmentally-friendly.  

General weaknesses: The Authors have studied only the microstructure of the produced materials. The paper would be much better if any other i.e. mechanical properties were investigated.

I strongly recommend to add to the paper the part with mechanical studies. If it is difficult, then the paper still may be accepted if the authors will refer to the following remarks and do the necessary corrections, which would significantly improve the paper:

  1. Major comments

- the Authors should studied or at least discuss the problem of interfacial bonding strength between the metal and ceramic in their composite. I am afraid that some of the carbon created during the production process may remain at the interface and reduce its mechanical properties. The following papers may be useful: DOI: 10.1016/j.apsusc.2016.12.130; DOI: 10.1016/j.compstruct.2018.06.071

  1. Minor comments

-  fonts in Figure 1 and 5 should be bigger.

Author Response

Dear Reviewers:
Thank you for your comments concerning our manuscript entitled “Realization of phase and microstructure control in Fe/Fe2SiO4-FeAl2O4 metal-ceramic by alternative microwave susceptors” (ID: materials-1562484). We thank the reviewer for the very helpful comments. We revised the manuscript according to the recommendations. We provide below with a brief “List of changes” in the revised manuscript, followed by our answers to the referee’s comments.

List of changes:

Comment 1: The Authors have studied only the microstructure of the produced materials. The paper would be much better if any other i.e. mechanical properties were investigated.

Response to Comment 1:

Our main starting point is to prepare waste tailings into ceramic materials by means of microwave treatment. However, the composition of tailings is complicated, and there are few reports on this in our country. So we have first explored the process based on iron ore concentrate and will subsequently extend to the utilization of tailings.

The main purpose of this article was written to represent the research of nanostructures in the course of process investigations, with a focus on changes in the susceptors that caused changes in the microstructure. Due to space limitations and the urgency of publishing this article, the suggestions made by the reviewers that address mechanical properties and AC residues will be explored in detail in subsequent articles.

In response to the reference provided by the reviewer, we are already discussing it. The reference is a significant reference for our research, and we will follow up with relevant research on our experimental materials.

Comment 2: fonts in Figures 1 and 5 should be bigger.

Response to Comment 2:

In response to the reviewer's suggestion that fonts in two figures in the text were small, we have made changes.

This is my first time writing an article in English, so I'm sure there are many shortcomings, and I hope the reviewers will criticize and correct me.

We thank the reviewer and remain at your disposal for any further questions.

Yours sincerely,

Chen-bo Gao

Round 2

Reviewer 2 Report

It seems, that the authors substantially corrected their manuscript, however there is still room for further improvement. Nevertheless, if the editor is satisfied with the revised version, I think it can be published.

Author Response

Dear Reviewer:
Thank you for your comments concerning our manuscript entitled “Realization of phase and microstructure control in Fe/Fe2SiO4-FeAl2O4 metal-ceramic by alternative microwave susceptors” (ID: materials-1562484). We thank the reviewer for the very helpful comments. We revised the manuscript according to the recommendations.

Thanks for your kind suggestion, we are very sorry for our poor English. Thank you for your technical review.

Yours sincerely,

Chen-bo Gao